# Investigation of the Vibrational Behavior of Thermoformed Magnetic Piezoelectrets

**DOI:** 10.3390/polym17111506

**Published:** 2025-05-28

**Authors:** Amélia M. Santos, Rui A. S. Moreira, Leonardo S. Caires, Ronaldo M. Lima, Elvio P. Silva, Polyane A. Santos, Jéssica F. Alves, Sergio M. O. Tavares, Kenedy Marconi G. Santos, Ruy A. P. Altafim, Ruy A. C. Altafim

**Affiliations:** 1TEMA—Centre for Mechanical Technology and Automation, Department of Mechanical Engineering, University of Aveiro, 3810-193 Aveiro, Portugalleonardocaires@ua.pt (L.S.C.); polyane@ua.pt (P.A.S.);; 2LASI—Intelligent Systems Associate Laboratory, 4800-058 Guimarães, Portugal; 3Department of Electrical Engineering, Federal Institute of Bahia, Vitoria da Conquista 45029-000, BA, Brazil; 4Department of Physics, Federal Institute of Sergipe, Lagarto 49400-000, SE, Brazil; 5Department of Electrical and Computer Engineering, University of São Paulo, São Carlos 13566-590, SP, Brazil; 6Computer System Department, Informatic Center, Federal University of Paraiba, João Pessoa 58055-000, PB, Brazil

**Keywords:** ferroelectrets, thermoformed piezoelectrets, magneto-electrets, polymers, piezoelectricity, laser vibrometer

## Abstract

This study explores the vibrational behavior of Thermoformed Magneto-Piezoelectrets (TMPs), multifunctional materials consisting of thermoformed piezoelectrets with open tubular channels integrated with an additional magnetic layer. The inverse piezoelectric effect was characterized using laser vibrometry analysis, measuring the mechanical response of TMPs subjected to electrical excitation over a frequency range of 0–20 kHz. Vibrational analysis was conducted at 144 spatial points, enabling the construction of detailed three-dimensional (3D) maps of the vibration operational modes and the spatial distribution of the piezoelectric coefficient ($d_{33}$). The results demonstrated significant frequency-dependent behavior, with open channels exhibiting pronounced resonance peaks, whereas valleys displayed smoother and more uniform responses due to enhanced damping effects. The observed heterogeneity in vibrational behavior is attributed to structural variations, material composition, and anisotropic coupling between the piezoelectric and magnetic properties. The findings presented in this research provide a comprehensive understanding of the development and utilization of TMPs, offering parameters for enhancing their application and supporting new discoveries in studies related to the fabrication of novel thermoformed piezoelectric sensors.

## 1. Introduction

Applications involving electromechanical effects frequently utilize piezoelectric materials, such as piezoceramics and electroactive polymers, including electrically charged polymer foams. The latter, known as ferroelectrets or piezoelectrets, are particularly notable for their ability to generate an electrical response and trap high levels of charge. Furthermore, their low density contributes to piezoelectric coefficients that exceed those observed in natural piezoelectric polymers [1,2].

Piezoelectrets have garnered considerable scientific interest due to their exceptional piezoelectric properties. These materials maintain electrical neutrality through a symmetric distribution of separated positive and negative charges. When subjected to mechanical stress, this symmetry is altered, leading to charge displacement and the generation of an electric potential throughout the material [3,4].

Advances in materials science have shown that chemical treatments or the combination of polymers with different elastic and thermal properties enhance charge-trapping efficiency and increase electromechanical sensitivity. These approaches have allowed the development of materials with piezoelectric coefficients that surpass even those of advanced ceramics [1,5,6].

Among the various polymers used, fluoropolymers such as polytetrafluoroethylene (PTFE) and its copolymer fluoroethylene propylene (FEP) are particularly effective in the creation of thermoformed piezoelectrets due to their superior thermal stability [7]. A notable design involving fluoropolymers is the formation of structures with open tubular channels. A methodology developed by Altafim [8] employs PTFE templates to create precisely aligned tubular cavities in thermally laminated FEP films. This technique enables customization of cavity geometries, resulting in organized multilayered structures that can further enhance piezoelectric performance [9,10,11].

Building on the combination of polymers with distinct properties, a novel concept of multifunctional material has been introduced: thermoformed magnetic-piezoelectrets (TMPs). These are multilayer electret structures based on piezoelectrets with open tubular channels augmented with an additional magnetic layer. This magnetic layer imparts the material with the ability to contract or expand the electrically charged tubular channels under the influence of an external magnetic field. This mechanical deformation disrupts the electrical equilibrium of the material, producing an electrical response proportional to the applied magnetic field [12].

Given that TMPs mechanically deform in response to an external magnetic field, they are hypothesized to exhibit a similar deformation when electrically stimulated. This study aims to investigate the vibrational behavior of TMPs to better understand their functional properties and explore their potential applications in sensor technology and energy harvesting.

## 2. Materials and Methods

This section describes the experimental procedures and methodologies adopted for the preparation, characterization, and vibrational analysis of TMPs. Using precise fabrication techniques and advanced measurement tools, the aim was to achieve a thorough understanding of the electromechanical properties and performance of the material. In the following, the fabrication of TMPs and the experimental setup for vibrational analysis are described in detail.

### 2.1. Preparation of Thermoformed Magnetic Piezoelectrets

The method detailed in [12] was used in the fabrication of the TMPs. Therefore, two FEP films, each 50 μm thick, were laminated together at 300 °C to form the multilayer structure. A 100 μm thick PTFE template, featuring rectangular cutouts, was placed between the FEP layers before lamination. This template guided the fusion process, creating structured channels. For this work, the template was produced by laser cutting and was designed to produce four parallel evenly spaced channels, each 2 mm wide and 30 mm long.

In the subsequent processing step, magnetic tape strips were applied to cover each channel. These strips were laser cut into rectangular shapes (1.5 mm × 15 mm) from a 300 μm thick laminated magnetic adhesive mat supplied by Fermag-BR. The addition of the magnetic layer introduced surface irregularities, which posed challenges to electrode formation. To address this issue, a third FEP film, also 50 μm thick, was laminated over the magnetic strips at 300 °C. Finally, the PTFE template was carefully removed, leaving open channels embedded within the FEP matrix. The entire production process is schematically illustrated in Figure 1.

After the layered structure was constructed, approximately 30 nm thick aluminum electrodes were deposited onto the external FEP layers through a vacuum deposition process, creating the conductive surfaces. The TMP devices were then electrically charged by applying a DC voltage of 3.5 kV for 10 s.

### 2.2. Vibrational Analysis

To perform the vibrational analysis of the TMP, an external electrical excitation was applied to stimulate the material. A white noise signal covering a frequency range from 0 to 20 kHz was generated by using a signal generator applied to the system. This signal was then amplified using a high voltage and high-speed linear amplifier (A.A. Lab Systems, model A-301), resulting in a final excitation amplitude of 160 V (320 $V_{pp}$). The amplified signal stimulated the TMP directly through its aluminum electrodes.

The mechanical response of the TMP to electrical excitation was measured using a laser vibrometer (Polytec, model OFV-505). Vibrational response data (point velocity) were collected from the vibrometer’s controller (Polytec, model OFV-5000—VD02) and further processed using a spectral analyzer (Spectral Dynamics, model SigLab 2042). In addition, a two-channel oscilloscope (GW Instek, model GDS-840C) was used for visual confirmation of the generated and amplified signals applied to the TMP and the velocity response signal provided by the laser vibrometer. A 100:1 oscilloscope probe was used to measure the amplified signal fed to the TMP.

The experimental setup is represented in Figure 2, where the components of the system are identified, including the excitation source, the measurement instruments, and the TMP sample.

The laser vibrometer was configured with the following specifications: a digital decover (VD02) with a sensitivity of 5 $\mathrm{mm}\;s^{-1}/V$, a frequency range of 0 to 20 kHz, and a low-pass filter adjusted to 20 kHz. The high-pass filter was disabled, and the tracking filter was set to fast mode for dynamic measurements. The TMP sample was placed in an acrylic receptacle mounted on an XY micrometric stage, enabling precise spatial control of the measurement points.

With the aim of conducting a detailed point-by-point characterization of the TMP, the sample was subdivided into 144 points and distributed in two types of columns: channels ($C_{a}$, $C_{b}$ and $C_{c}$) with a width of 2 mm, and valleys ($V_{1}$, $V_{2}$ and $V_{3}$) with a width of 3 mm. A fixed vertical spacing of 1 mm was adopted between the rows. Figure 3a illustrates the TMP on the scale. The representation of the channels and valleys is shown in Figure 3b. Figure 3c presents the spatial position of the 144 points measured in the experiment.

Channels correspond to regions of the TMP that contain open tubular air, whereas valleys represent areas where the material fused during the thermoforming process and are, therefore, expected to exhibit a reduced or negligible vibration response. The laser from the vibrometer was precisely positioned at each point during the experiment using the XY micrometric stage, ensuring high positional accuracy. For each measurement position, multiple readings were acquired, with an average of at least 100 time samples per point to ensure reliability and precision.

## 3. Results and Discussion

At each reading point, mobility frequency response functions (magnitude and phase) were measured over the range of 0 Hz to 20 kHz and subsequently imported into MATLAB for analysis.

The $d_{33}$, which quantifies the mechanical deformation of the material in response to the applied electric field, is obtained by using the magnitude of the displacement measurement and the magnitude of the excitation voltage. The excitation voltage $V_{ex}$ is represented as a sinusoidal function:(1)$$V_{ex}=V_{max}\cdot sin\left(\omega t\right),$$
where $V_{max}$ is the maximum excitation voltage, and ω is the angular frequency of the excitation. The measured displacement $\mu_{out}$ is represented as:(2)$$\mu_{out}=\mu_{max}\cdot sin\left(\omega t+\phi \right),$$
where $\mu_{max}$ is the displacement amplitude, and ϕ is the phase shift relative to the excitation signal. $d_{33}$ is calculated as expressed in Equation (Equation 3):(3)$$d_{33}=\frac{\mu_{max}}{V_{max}}.$$

The direct output is a voltage signal from the vibrometer velocity decoder. The displacement amplitude is calculated by using the voltage output and sensitivity of the velocity decoder:(4)$$\mu_{max}=\frac{V_{amp}\cdot \delta }{\omega },$$
where $V_{amp}$ is the amplitude voltage output from the velocity decoder, and δ is the decoder sensitivity. The ω denominator, representing the angular frequency, performs the signal integration from velocity to displacement in the frequency domain.

The three-dimensional surface map highlights variations in the $d_{33}$ values across the TMP surface at a frequency of 50 Hz in Figure 4. Peaks on the surface correspond to regions with enhanced piezoelectric response, while valleys represent areas of reduced activity. The top-view map offers a complementary two-dimensional perspective, emphasizing the spatial heterogeneity of the piezoelectric response.

This non-uniform distribution is indicative of structural and compositional differences within the TMP, likely caused by variations in molecular alignment, magnetic particle distribution, and residual stresses introduced during fabrication. The heterogeneity in $d_{33}$ values can also be attributed to anisotropic coupling between the magnetic and piezoelectric properties of the material. In anisotropic systems, the interaction between these properties varies depending on spatial direction, resulting from structural inhomogeneities at the microscopic level.

Differential alignment of molecular chains or embedded magnetic particles during manufacturing likely contributes to the directional dependency of the piezoelectric response. Furthermore, non-uniform particle distribution and local charge accumulation could amplify these effects, while residual stresses generated during processing might further influence the observed anisotropy.

The observed heterogeneity in the $d_{33}$ distribution can be attributed to multiple physical factors. Structurally, the open channels exhibit different local stiffness compared to the thermoformed valleys, directly affecting the mechanical deformation modes under electrical excitation. Furthermore, the thermal lamination process may induce preferential orientation of polymer chains in materials such as FEP, leading to local anisotropies in the piezoelectric response [1,4].

The magnetic layer, in turn, introduces a non-uniform mass distribution across the TMP surface, locally affecting the eigenmodes of vibration. Small variations in thickness, adhesion, and density of this layer impact the mechanical stiffness and resonance behavior of the structure. Finally, anisotropic coupling between magnetic and piezoelectric properties may arise from the spatial organization of the magnetic layer and its interaction with the internal structure of the cavities [12], even in the absence of an external magnetic field. These factors act synergistically, modulating the local sensitivity to electrical excitation and resulting in the spatial variations of $d_{33}$ observed experimentally.

Figure 5 provides a detailed visualization of the $d_{33}$ distribution at a frequency of 5.8 kHz, showing both the three-dimensional surface map (a) and the corresponding two-dimensional top view (b).

Unlike the data obtained at lower frequencies, the measurements at 5.8 kHz reveal finer spatial variations in the piezoelectric response, with more pronounced peaks and valleys across the surface. These results indicate that frequency-dependent effects may play a significant role in enhancing or suppressing the piezoelectric response in certain regions, likely as a result of resonance phenomena or frequency-specific material behaviors.

The top view in Figure 5b further highlights the periodicity and anisotropy in the spatial distribution of the values of $d_{33}$. The observed patterns suggest localized enhancements in magneto-piezoelectric coupling at specific positions, potentially linked to structural or compositional factors. At this frequency, the coupling mechanisms may interact more efficiently with the intrinsic properties of the polymer, leading to distinct regions of higher activity.

Figure 6 illustrates the spatial distribution of the inverse piezoelectric effect at a frequency of 12 kHz. At this frequency, the surface exhibits more pronounced peaks and valleys compared to lower frequencies, indicating a greater degree of localized piezoelectric activity. Figure 6b presents a top-view map, highlighting a more consistent distribution of piezoelectric behavior across the material at higher frequencies.

The spatial distribution of $d_{33}$ at a frequency of 20 kHz is shown in Figure 7, showcasing unique features compared to previous measurements. In Figure 7a, the 3D surface map reveals that the piezoelectric response exhibits sharper and more localized variations at this frequency. This could indicate an increased sensitivity of the material to higher-frequency electric fields, potentially due to specific resonance effects or the dynamic alignment of its microstructural domains.

The sharper features may also highlight regions with enhanced piezoelectric coupling, further influenced by the frequency-dependent interaction of the material’s properties. In Figure 7b, the top view map offers greater contrast between regions with high and low $d_{33}$ values.

This enhanced contrast suggests a more defined spatial pattern at 20 kHz, potentially indicative of frequency-induced structural polarization or alignment effects within the material. The observed distribution also underscores the importance of frequency in modulating the piezoelectric response across the surface, with certain regions becoming more prominent in their activity.

This analysis reveals the significant frequency-dependent behavior of the TMP, as evidenced by the spatial distribution of the $d_{33}$ across the tested frequency range. At lower frequencies, the piezoelectric response exhibited a more uniform and less pronounced distribution, while at higher frequencies, such as 12 kHz and 20 kHz, sharper and more localized variations in $d_{33}$ values were observed. These results suggest the presence of resonance effects or frequency-specific alignment of the material’s domains, enhancing piezoelectric performance in particular regions.

The irregular vibration behavior observed in the mapping of the TMP structures can be partially explained by the non-uniform distribution of electric charges on the internal surfaces of the cavities. Although the tubular geometry of the TMP channels maintains a degree of consistency and regularity, this variation, combined with the intensification of the electric field at the edges of the channel, results in different modes of mechanical deformation along the channels. Additionally, the production process of the TMP contributes to these multiple vibration modes. Each channel is covered with an extra layer of magnetic material, adding weight and further distorting the structure, which amplifies these irregularities.

The magnetic layer was incorporated to provide multifunctionality to the TMPs by enabling magnetoelectric coupling. In the present study, no external magnetic field was applied; instead, the magnetic layer primarily acts as a localized mass, altering the vibrational modes and influencing the distribution of $d_{33}$. In [12,13], the interaction between external magnetic fields and cavity deformation was actively investigated to quantify the magneto-piezoelectric effect in TMPs.

Further analysis was conducted on the gain curves obtained during vibrometer testing. The figures below present 12 gain curves processed using a 10-point moving average filter, providing a clearer visualization of the data. Each line corresponds to a specific measurement point, highlighting the distinct responses observed within the TMP’s structured channels and valleys. Figure 8a specifically illustrates the $d_{33}$ curves for the central points in channel A.

In channel A, the peaks in the curves correspond to the resonance frequencies, where the piezoelectric response reaches its maximum, indicating optimal efficiency in converting electrical energy into mechanical deformation. Sharp and narrow peaks indicate low damping, where vibrational energy is efficiently retained, resulting in high sensitivity and a narrow operational bandwidth. In contrast, broader peaks suggest higher damping, with energy dissipating more rapidly, leading to a wider bandwidth but reduced sensitivity. Figure 8b displays the $d_{33}$ curves for the central points in valley 1.

Compared to the central points in channel A, the curves in valley 1 generally exhibit lower $d_{33}$ values, indicating a reduced piezoelectric response in this region. This is consistent with the structural geometry of the TMP, where valleys tend to experience less mechanical deformation due to the distribution of stress and electric fields being concentrated in the channels.

The curves in valley 1 demonstrate less pronounced resonance peaks, suggesting that the piezoelectric activity in these regions is less sensitive to specific frequencies. This behavior indicates higher damping effects in the valleys, as energy is dissipated more uniformly, and the system does not strongly favor certain frequencies for resonance. Additionally, the response across the valley points appears more homogeneous compared to the channels, as evidenced by the reduced variability between the curves. This homogeneity can be attributed to the relatively uniform structural and material properties within the valley regions.

At higher frequencies beyond 15 kHz, the $d_{33}$ values show a consistent decline, reflecting the transition to a regime dominated by damping. The lack of sharp resonance features in this region highlights the diminished capacity of the valleys to support dynamic responses compared to the channels.

These observations suggest that the valleys contribute less to the overall piezoelectric performance of the TMP and may be less critical for applications that require high sensitivity or resonance tuning. Figure 8c presents the $d_{33}$ curves for the central points in channel B.

Channel B exhibits greater variability in amplitude across the measured points, suggesting a less uniform distribution of piezoelectric activity. While resonance peaks are present, they display greater irregularities in both shape and frequency position than those observed in channel A. At low frequencies, the $d_{33}$ values for channel B are relatively stable; however, as the frequency increases beyond approximately 10 kHz, the variability in the response becomes more pronounced. The decay of $d_{33}$ values at higher frequencies is less consistent compared to other regions, with some points maintaining higher amplitudes than others. This could indicate that certain areas within channel B retain energy more effectively, possibly due to localized resonant effects or reduced damping in specific sections. Figure 8d presents the $d_{33}$ curves for the central points in valley 2.

Unlike other regions analyzed, valley 2 demonstrates a notable consistency in the amplitude of $d_{33}$ values across different points, suggesting a relatively uniform piezoelectric response. This homogeneity could be attributed to the structural features of valley 2, where stress and charge distribution are more evenly spread compared to the channels or other valleys.

Although resonance peaks are present, their amplitudes are less pronounced, and the overall frequency-dependent response appears smoother than in regions such as channel B. The smoother behavior across frequencies suggests that valley 2 dissipates vibrational energy more evenly, which may correlate with higher damping properties compared to regions with sharper resonance features.

Interestingly, the $d_{33}$ values remain relatively stable at mid frequencies (around 5–15 kHz), showing less fluctuation compared to the other valleys or channels. This stability implies that valley 2 contributes to a steadier mechanical response over this frequency range, potentially enhancing the reliability of the TMP in applications where consistent behavior across frequencies is required.

At higher frequencies (above 15 kHz), the decay in $d_{33}$ values is gradual, maintaining a more predictable decline compared to other regions. This indicates that valley 2 retains a degree of responsiveness at frequencies where other areas may show significant reductions. Figure 8e illustrates the $d_{33}$ curves for the central points in channel C.

Compared to previous regions analyzed, channel C exhibits a moderate degree of variability in amplitude across the points, suggesting localized differences in piezoelectric activity. However, the response within channel C shows more intermediate behavior, balancing some of the pronounced characteristics observed in channels A and B.

The resonance peaks in channel C are generally less sharp, with amplitudes that are moderate compared to channels with lower damping or stronger localized coupling. This indicates that the mechanical–electrical coupling in channel C may be influenced by both structural and material properties that moderate extreme resonance behavior. The absence of highly distinct resonance frequencies suggests that channel C is less prone to localized resonances and operates within a more damped regime compared to regions with sharper peaks.

At low frequencies, the $d_{33}$ values in channel C exhibit more noticeable fluctuations than other regions, indicating that this channel may be more sensitive to low-frequency dynamics. In the higher frequency range (above 15 kHz), the decay of the values $d_{33}$ is relatively smooth and consistent at most points. This reflects a more predictable damping behavior compared to regions where sharper declines were observed. Such stability at high frequencies may indicate that channel C offers more reliable performance under dynamic conditions where high-frequency responses are critical. Figure 8f illustrates the $d_{33}$ curves for the central points in valley 3.

The behavior observed in valley 3 is characterized by moderate variability among the measurement points, indicating a slightly less uniform piezoelectric response compared to valley 2. However, the variations remain more subdued than those observed in the channels, suggesting that valley 3 retains some degree of structural homogeneity.

The resonance peaks in valley 3 are generally less pronounced, with smoother transitions between frequencies. This indicates a relatively damped behavior, where energy dissipation is more uniform and localized resonances are less dominant. The reduced prominence of resonance features in this valley suggests that the mechanical–electrical coupling is less sensitive to specific frequencies compared to regions with sharper peaks.

Interestingly, the mid-frequency range (approximately 5–10 kHz) shows slightly more pronounced fluctuations compared to other valleys, suggesting that valley 3 may have localized structural or material variations that impact its response in this range. These mid-frequency features could be indicative of subtle resonance effects influenced by the valley’s geometry or stress distribution.

Previous studies have demonstrated the potential of piezoelectrets with open cavities for sensor and actuator applications, particularly due to their high sensitivity, flexibility, and low density [1,10,11]. Works such as those by Altafim et al. [12] have advanced the characterization of piezomagnetic structures by analyzing their behavior under external magnetic fields. However, most of these investigations focus on global measurements of the piezoelectric effect or static analyses without thoroughly exploring the local vibrational response and the spatial distribution of the $d_{33}$ coefficient. This study makes an original contribution by performing a spatially resolved vibrational analysis of TMPs, using laser vibrometry to map 144 points across different structural regions. This enables observation of the dynamic behavior of thermoformed cavities under electrical excitation, providing new experimental parameters for the geometric and functional optimization of TMP-based devices.

At higher frequencies beyond 15 kHz, the $d_{33}$ values in valley 3 exhibit a consistent and gradual decline similar to valley 2. This predictable behavior at higher frequencies reinforces the role of valleys in providing steadier and less dynamic responses compared to channels. However, the slightly greater variability among the points in valley 3 suggests that it may be more susceptible to localized factors that affect its performance.

Valley 3 demonstrates characteristics that balance damping and variability, with slightly more dynamic mid-frequency behavior than other valleys. This suggests that valley 3 could contribute to applications that require moderate piezoelectric activity across a range of frequencies while maintaining sufficient damping to avoid excessive resonant effects. These findings highlight the nuanced differences between valleys and their roles in shaping the overall performance of the TMP.

These results reveal distinct piezoelectric behaviors across the channels and valleys of the TMP, emphasizing the influence of structural geometry and material properties on the piezoelectric response. Channels generally display more pronounced resonance peaks and greater variability, reflecting localized sensitivity and structural heterogeneity. In contrast, valleys exhibit smoother and more uniform responses, with reduced resonance activity and higher damping, indicating their role in providing stability and efficient energy dissipation.

Although this study focuses on the experimental investigation of the vibrational behavior of TMPs, the authors acknowledge the importance of a numerical model to predict the spatial distribution of the piezoelectric coefficient, $d_{33}$. The variation observed in the experimental maps can be qualitatively attributed to the interaction among the geometry of the open channels, charge distribution, and the anisotropic coupling between magnetic and piezoelectric properties. Future work will involve the implementation of finite element analysis (FEA) to quantitatively validate the experimental results and further explore local resonant effects and anisotropic coupling mechanisms.

## 4. Conclusions

Thermoformed Magnetic Piezoelectrets (TMPs) represent an innovative class of multifunctional materials that combine piezoelectric and magnetoelectric properties, enabling dual responsiveness to mechanical and magnetic stimuli. This study investigated the vibrational behavior of TMPs with structured open channels and a magnetic layer, emphasizing the inverse piezoelectric effect, where electrical excitation induces mechanical deformation. Using a laser vibrometer, vibrational responses were mapped at 144 spatial points throughout a frequency range of 0–20 kHz, allowing the creation of detailed 3D vibration profiles.

The findings reveal that TMP channels exhibit prominent resonance peaks and variability in the values of $d_{33}$, indicating high sensitivity and localized piezoelectric activity and structural heterogeneity. In contrast, valleys demonstrate smoother and more uniform responses with increased damping, contributing to stable energy dissipation. These contrasting behaviors suggest that channels are better suited for sensitivity-driven applications, whereas valleys are more appropriate for scenarios that require reliability and uniform performance.

The integration of a magnetic layer enhances the multifunctionality of TMPs by introducing magnetoelectromechanical coupling, enabling applications in sensors, energy harvesting, and actuators. The observed inverse piezoelectric effect highlights their efficiency over a broad frequency range, offering essential perspectives into their dynamic capabilities.

This study provides a deeper understanding of the vibrational characteristics of TMPs, emphasizing the interplay among structural geometry, material properties, and external stimuli. The detailed mapping of $d_{33}$ values across the TMP surface provides a foundation for optimizing its design, enabling customized improvements for specific applications with required structural and functional properties. Future work could explore strategies to enhance material uniformity, refine manufacturing techniques, and further investigate the coupling between piezoelectric and magnetoelectric effects, paving the way for broader adoption of TMPs in advanced technological applications.

The results obtained in this study provide direct insight into improving the design of TMPs in practical applications. Identification of regions with intensified piezoelectric response, associated with open channels and resonant frequencies, enables the optimization of transducer geometric layouts for applications in high-sensitivity vibration sensors, electric current detection in embedded systems, and energy harvesting in portable devices.

Future work will include the exploration of different polymer compositions, especially materials with enhanced thermal stability and greater charge retention capability, as well as the use of magnetic layers with tailored properties to reinforce magnetoelectric coupling. Furthermore, the lamination and cavity formation processes will be refined to improve the reproducibility of the piezoelectric response and minimize the spatial variability of $d_{33}$, targeting applications demanding uniform response, such as distributed sensor networks or implantable biomedical devices.

Furthermore, the data obtained in this study may guide the development of devices with tuned resonance, where the geometry and arrangement of the cavities are designed to maximize the piezoelectric response at specific frequency ranges. Future investigations will consider the use of alternative geometries and the introduction of new functional materials, aiming at the development of micro-positioning actuators and intelligent interfaces for magnetic detection in dynamic environments.

## Figures and Tables

**Figure 1 polymers-17-01506-f001:**
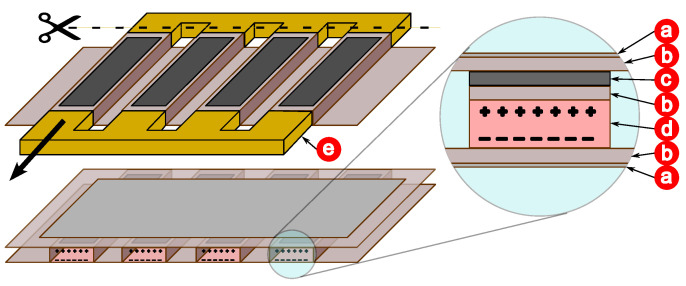
Fabrication procedure for TMPs with open tubular channels and piezoelectric-magnetic response: (**a**) electrode, (**b**) FEP, (**c**) magnectic layer, (**d**) air cavity, and (**e**) PTFE.

**Figure 2 polymers-17-01506-f002:**
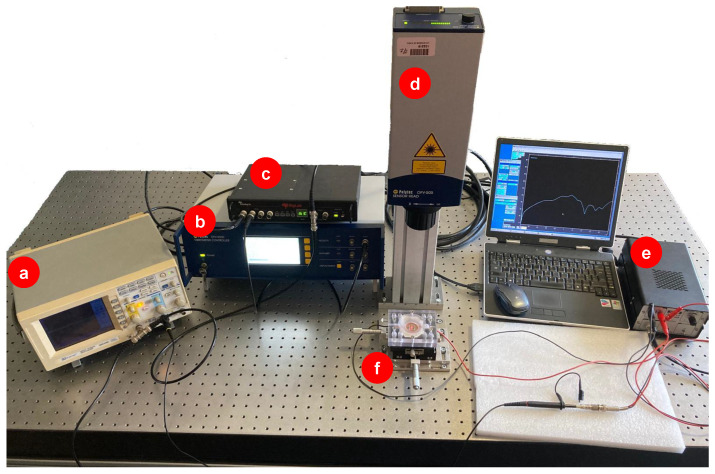
Experimental setup: (**a**) oscilloscope, (**b**) vibrometer controller, (**c**) dynamic signal analyzer, (**d**) laser vibrometer, (**e**) high voltage amplifier, and (**f**) TMP.

**Figure 3 polymers-17-01506-f003:**
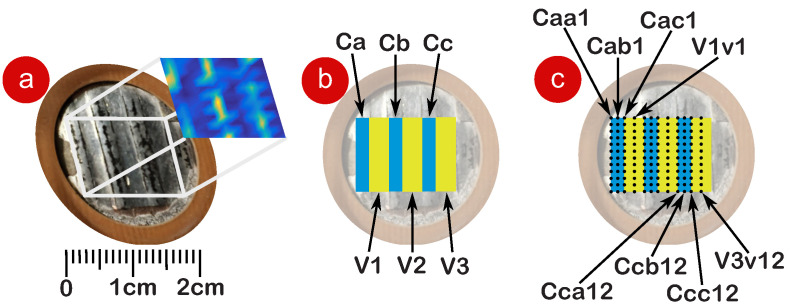
Measurement methodology. (**a**) Scaled TMP, (**b**) representation of the channels (Ca, Cb, and Cc) and valleys (V1, V2, and V3), and (**c**) positioning of the 144 laser measurement points.

**Figure 4 polymers-17-01506-f004:**
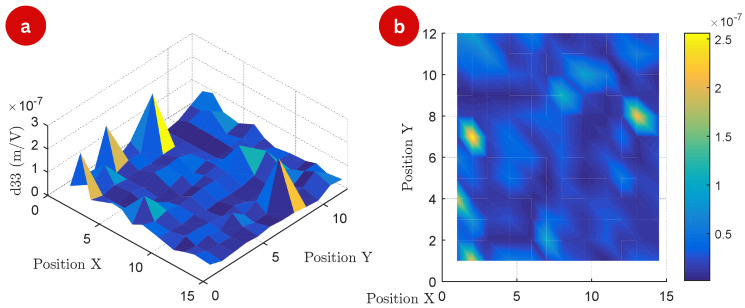
Surface map of the experiment: (**a**) 3D map of the measured $d_{33}$ matrix (m/V) of the 144 measurement points at a frequency of 50 Hz, (**b**) top view of the matrix points.

**Figure 5 polymers-17-01506-f005:**
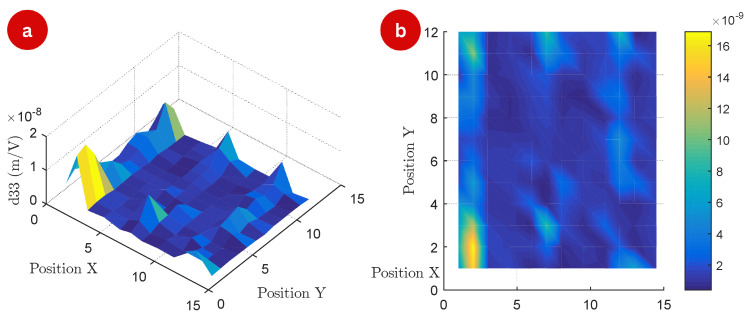
Surface map of the experiment: (**a**) 3D map of the measured $d_{33}$ matrix (m/V) of the 144 measurement points at a frequency of 5.8 kHz, and (**b**) top view of the matrix points.

**Figure 6 polymers-17-01506-f006:**
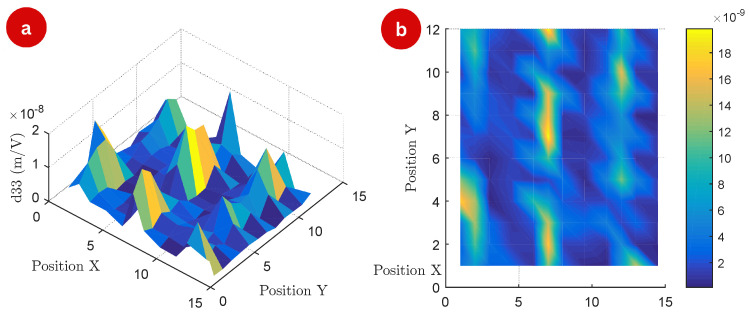
Surface map of the experiment: (**a**) 3D map of the measured $d_{33}$ matrix (m/V) of the 144 measurement points at a frequency of 12 kHz, and (**b**) top view of the matrix points.

**Figure 7 polymers-17-01506-f007:**
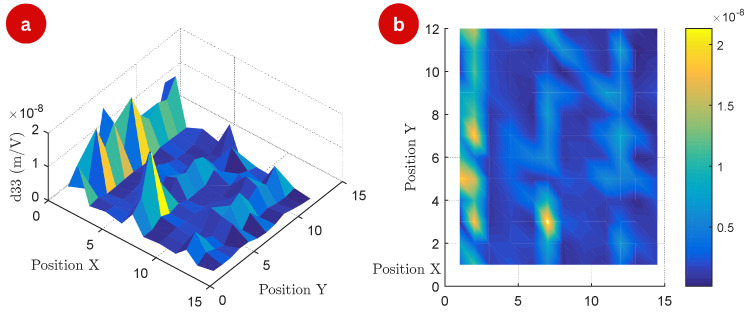
Surface map of the experiment: (**a**) 3D map of the measured $d_{33}$ matrix (m/V) of the 144 measurement points at a frequency of 20 kHz, and (**b**) top view of the matrix points.

**Figure 8 polymers-17-01506-f008:**
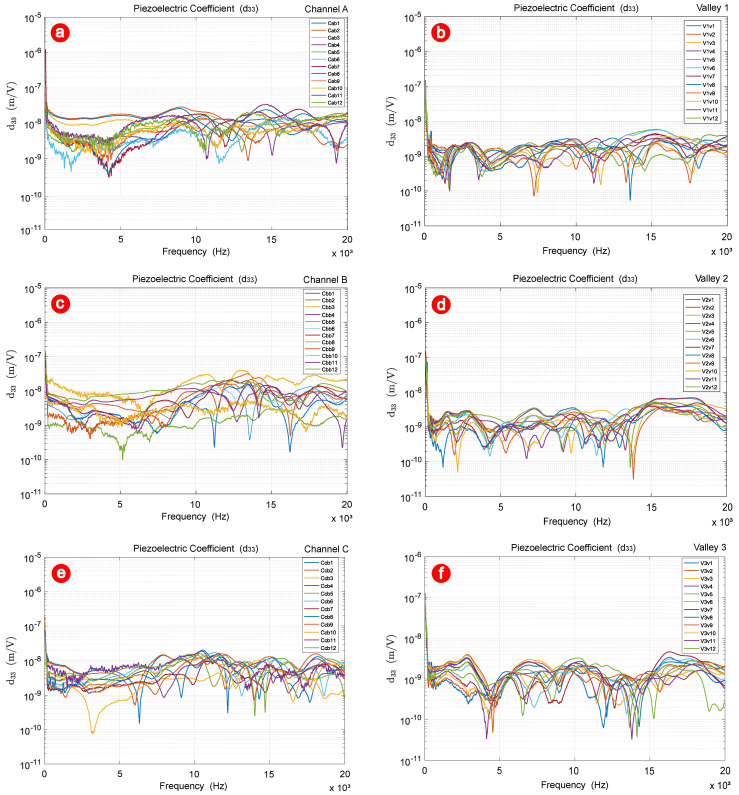
Piezoelectric coefficient curves of the central points: (**a**) channel A; (**b**) valley 1; (**c**) channel B; (**d**) valley 2; (**e**) channel C; (**f**) valley 3.

## Data Availability

The original contributions presented in the study are available at: https://doi.org/10.5281/zenodo.15318794. Further inquiries can be directed to the corresponding author.

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
