# Peer review of "Investigation of the Vibrational Behavior of Thermoformed Magnetic Piezoelectrets"

_polymers, 2025, doi:10.3390/polym17111506_

Round 1

Reviewer 1 Report

Comments and Suggestions for Authors

This manuscript reports an experimental investigation of the vibrational behavior of thermoformed magneto-piezoelectrets (TMPs). The key novelty of this work appears to be the structural design of TMPs. The channel and valley structures in TMPs enable the significant difference of frequency-dependent vibration behaviors between the two structures. The different vibration behaviors have been experimentally investigated from the spatial distribution of the piezoelectric coefficient at a spatial resolution of 144 points. However, the physical mechanism behind the experimental results seems unclear, and no reliable explanation has been given for the role of each component in TMPs. Therefore, this paper requires significant revision before it can be considered for publication. Here are some technical suggestions:

  1. More details of device fabrication should be provided. For example, the sizes of each FEP film and air cavity, PTFE template dimensions, alignment methods, and magnetic layer properties (e.g., composition, thickness, magnetization).
  2. I recommend the authors to add a theoretical model (e.g., finite element analysis) to predict d33 distribution and validate experimental maps.
  3. The roles of the components in TMPs should be experimentally explained. For example, how does the thickness of FEP film affect the d33 and vibration behaviors?
  4. What is the role of the magnetic layer of TMPs? The authors did not introduce any magnetic field in this work. How does the magnetic layer work?
  5. The authors mentioned that the TMP devices needed a very high excitation voltage of 320 V. Why? And how to reduce the excitation voltage since it is too high for actual applications.
  6. The letter ω represents two parameters (the angular frequency of the excitation and signal integration from velocity to displacement) in the manuscripts. It might cause confusion.

Author Response

This manuscript reports an experimental investigation on the vibrational behavior of thermoformed magneto-piezoelectric materials (TMPs). The main novelty of the work appears to be the structural design of the TMPs. The channel and valley structures allow for a significant difference in frequency-dependent vibrational behaviors. The different vibrational responses were experimentally studied based on the spatial distribution of the piezoelectric coefficient, with a spatial resolution of 144 points. However, the physical mechanisms behind the experimental results remain unclear, and no reliable explanation has been provided for the role of each component in the TMPs. Therefore, this article requires major revisions before it can be considered for publication. Here are some technical suggestions:

1- More details about the device fabrication should be provided. For example, the sizes of each FEP film and air cavity, PTFE mold dimensions, alignment methods, and properties of the magnetic layer (e.g., composition, thickness, magnetization).

Response: We appreciate the reviewer’s comment. The fabrication process of the TMPs involves preparing a polytetrafluoroethylene (PTFE) mold, which is placed between two fluorinated ethylene propylene (FEP) films to form tubular cavities through a thermoforming process. A 100 µm thick PTFE sheet is laser-cut into rectangular patterns (2 mm width × 30 mm length) to serve as spacers during the FEP fusion. After cleaning the 50 µm thick FEP films and the mold with acetone, the multilayer structure is inserted into a thermal laminator preheated to 300°C. Following the fusion of the FEP layers at the designated areas, a magnetic layer is added. This layer consists of a 300 µm thick adhesive magnetic tape (supplied by Fermag-BR), laser-cut into strips of 1.5 × 15 mm. The magnetic strips are aligned with the non-fused FEP regions that will later form the channels. A third 50 µm FEP film is then placed on top, and an additional lamination step at 300°C is performed to smooth the surface of the TMP. After this process, the PTFE mold is removed, creating the tubular structures with the magnetic strips aligned on top. Finally, 30 nm thick aluminum electrodes are deposited on both surfaces of the TMP for electrical charging and measurements.

Additional information was included in Subsection 2.1.

“The method detailed in [ 12 ] was used in the fabrication of the TMPs. Therefore, two FEP films, each 50 μm thick, were laminated together at 300â—¦C to form the multilayer structure. A 100 μm thick PTFE template, featuring rectangular cutouts, was placed between the FEP layers before lamination. This template guided the fusion process, creating structured channels. For this work, the template was produced by laser cut and was designed to produce four parallel evenly spaced channels, each 2 mm wide and 30mm long.

In the subsequent processing step, magnetic tape strips were applied to cover each channel. These strips were laser cut into rectangular shapes (1.5 mm × 15 mm) from a 300 μm thick laminated magnetic adhesive mat supplied by Fermag-BR. The addition of the magnetic layer introduced surface irregularities, which posed challenges to electrode formation. To address this issue, a third FEP film, also 50 μm thick, was laminated over the magnetic strips at 300â—¦C. Finally, the PTFE template was carefully removed, leaving open channels embedded within the FEP matrix. The entire production process is schematically illustrated in Figure 1.

After the layered structure was constructed, approximately 30nm thick aluminum electrodes were deposited onto the external FEP layers through a vacuum deposition process, creating the conductive surfaces. The TMP devices were then electrically charged by applying a DC voltage of 3.5 kV for 10 seconds.”

2- The authors are encouraged to include a theoretical model (e.g., finite element analysis) to predict the d33 distribution and validate the experimental maps.

Response: We thank the reviewer for this insightful suggestion. We agree that finite element analysis (FEA) could serve as an important complementary tool for validating the experimental data presented in this study. However, the primary focus of this manuscript is on high-resolution experimental characterization of the vibrational response of TMPs. A comprehensive computational model is currently being addressed by the research group and will be disclosed in a dedicated publication focused on the validation and exploration of resonance effects and anisotropic coupling.

For reference, a combined experimental and FEA study on charge deposition in tubular ferroelectrets is available in:

Neerajan Nepal, Ruy Alberto Pisani Altafim, Axel Mellinger; Space charge deposition in tubular channel ferroelectrets: A combined fluorescence imaging/LIMM study with finite element analysis. J. Appl. Phys., June 28, 2017; 121 (24): 244103. https://doi.org/10.1063/1.4990280

To partially address the reviewer’s recommendation, we have included a qualitative explanation of the physical mechanisms underlying the observed d33 distribution, as well as a paragraph in Section 3 outlining the future implementation of FEA simulations.

Added text in Section 3:

"Although this study focuses on the experimental investigation of the vibrational behavior of TMPs, the authors acknowledge the importance of numerical modeling for predicting the spatial distribution of the piezoelectric coefficient d33. The variation observed in the experimental maps can be qualitatively explained by the interplay between the geometry of the open channels, charge distribution, and anisotropic coupling between the magnetic and piezoelectric properties. Future work will include the implementation of finite element simulations to quantitatively validate the experimental results and explore resonance effects and local coupling in greater detail."

3- The roles of the TMP components should be experimentally clarified. For example, how does the thickness of the FEP film affect the d33 and vibrational behavior?

Response: This study employed a fixed FEP film thickness of 50 µm. We acknowledge that variations in FEP thickness directly influence dielectric stiffness, electric field distribution, and mechanical coupling, ultimately affecting piezoelectric sensitivity (d33) and vibrational modes. A systematic parametric experimental study is planned for future work to investigate the impact of FEP thickness on vibrational behavior and piezoelectric coupling efficiency. For further insights on this topic, we recommend the following article:

Zhukov, S., Eder-Goy, D., Fedosov, S. et al. Analytical prediction of the piezoelectric d33 response of fluoropolymer arrays with tubular air channels. Sci Rep 8, 4597 (2018). https://doi.org/10.1038/s41598-018-22918-1

4- What is the role of the magnetic layer in the TMPs? The authors did not apply any magnetic field in this study. How does the magnetic layer function?

Response: We thank the reviewer for this important question. The magnetic layer was introduced to provide multifunctionality to the TMPs, enabling magnetoelectric coupling in future applications. Although no external magnetic field was applied in this study, the presence of the magnetic layer significantly affects the vibrational response by acting as a localized additional mass that alters the vibrational modes and the spatial distribution of the piezoelectric coefficient d33.

The active investigation of magnetoelectric effects under applied magnetic fields has been previously conducted by our team in studies such as:

  1. M. Santos et al., "Current Transducer Based on Thermoformed Piezo-Magnetic-Electrets," IEEE Sensors Journal, 2024.

  1. A. P. Altafim et al., "Piezoelectric-magnetic behavior of ferroelectrets coated with magnetic layer," Applied Physics Letters, 2021.

To clarify this aspect, we added the following explanation at the end of Section 3:

"The magnetic layer was incorporated to enable multifunctionality in the TMPs by allowing magnetoelectric coupling. In this study, no external magnetic field was applied; the magnetic layer primarily acts as a localized mass that alters the vibrational modes and influences the spatial distribution of d33. Previous studies [1,2] have actively investigated the interaction between external magnetic fields and cavity deformation to quantify the magneto-piezoelectric effect in TMPs."

5- The authors mention that the TMPs require a high excitation voltage of 320 V. Why is this necessary, and how can the excitation voltage be reduced, given that this is too high for practical applications?

Response: Thank you for this observation. We clarify that the voltage reported in the manuscript refers to a peak-to-peak value of 320 V (160 V amplitude), used exclusively for experimental purposes to ensure a sufficiently strong vibrational signal detectable by the laser vibrometer. This value is not an intrinsic requirement of the TMPs but was chosen to maximize the signal-to-noise ratio during characterization, given the low density and high damping of the structures. While we agree that this voltage is high for certain electronic applications, it remains within acceptable ranges for applications such as power sensors.

6- The symbol ω represents two parameters (the angular excitation frequency and the integration factor for velocity-to-displacement conversion) in the manuscript, which may cause confusion.

Response: We acknowledge the reviewer’s remark as the use of the angular frequency ω may raise some misinterpretation. In fact, the use of the angular frequency in Equation (4) is correct since:

  • The functions being used in this analysis are described in frequency domain
  • To simplify the analysis, only the magnitude of the frequency domain functions is analyzed
  • In frequency domain, derivation of a harmonic signal is performed by multiplying by jω (where j is the imaginary unit) and its integration is obtained by dividing it by jω.
  • Since the analysis is made solely on the magnitude (phase effects are disregarded), the magnitude of the integrated signal (displacement) is then the velocity magnitude divided by ω.

Text was improved to solve this issue.

Reviewer 2 Report

Comments and Suggestions for Authors

In this manuscript, the authors investigate the vibrational behavior of Thermoformed Magnetic Piezoelectrets (TMPs) using laser vibrometry analysis. The study provides detailed 3D maps of the vibration operational modes and the spatial distribution of the piezoelectric coefficient (d33) over a frequency range of 0–20 kHz. The manuscript is well-structured, rich in data, and the results are clearly presented. However, there are still some aspects that need further clarification or improvement. Based on the considerations above, this manuscript should be accepted after minor modification:

  1. The manuscript mentions that the observed heterogeneity in vibrational behavior is attributed to structural variations, material composition, and anisotropic coupling between the piezoelectric and magnetic properties. However, a more detailed explanation of how these factors specifically influence the vibrational behavior is lacking. For example, how do the molecular alignments or magnetic particle distributions affect the d33 values in different regions? A more in-depth discussion on the underlying mechanisms would enhance the understanding of the results. 2. The magnetic layer plays a crucial role in the functionality of TMPs. However, the manuscript does not provide detailed characterization of the magnetic layer’s properties, such as its magnetic susceptibility, coercivity, or remanence. It is essential to include these characterizations to fully understand the magnetoelectric coupling in TMPs and its impact on the vibrational behavior. 3. The results show significant frequency-dependent behavior of the piezoelectric coefficient (d33). At higher frequencies, sharper and more localized variations in d33 values are observed. The authors should provide a more comprehensive explanation for this frequency-dependent behavior. For example, are there specific resonance modes or dynamic alignment effects that dominate at higher frequencies? A detailed discussion on the possible mechanisms behind these observations would be beneficial. 4. The manuscript highlights the non-uniform distribution of d33 values across the TMP surface. While this is an important finding, it would be helpful to include a quantitative analysis of the uniformity or variability of d33 values. For instance, calculating the standard deviation or coefficient of variation of d33 values across different regions (channels and valleys) could provide a clearer picture of the material’s consistency. This information is crucial for optimizing the TMPs for specific applications. 5. The introduction mentions previous studies on piezoelectrets and their applications. However, there is limited discussion on how this study compares with or builds upon existing research. The authors should provide a more detailed comparison with previous works, highlighting the novelty and significance of their findings. This would help to contextualize the study within the broader field of piezoelectric materials. 6. The conclusion section briefly mentions potential applications of TMPs in sensor technology and energy harvesting. However, a more detailed outlook on future work and specific applications would strengthen the manuscript. The authors should discuss how the findings from this study could be leveraged to improve the design and performance of TMPs for practical applications. Additionally, suggestions for future research directions, such as exploring different material compositions or optimizing the fabrication process, would be valuable.

Author Response

In this manuscript, the authors investigate the vibrational behavior of Thermoformed Magnetic Piezoelectrets (TMPs) using laser vibrometry analysis. The study provides detailed 3D maps of the operational vibration modes and the spatial distribution of the piezoelectric coefficient (d33) across a frequency range of 0–20 kHz. The manuscript is well-structured, rich in data, and the results are clearly presented. However, there are still some aspects that require further clarification or improvement. Based on the considerations above, this manuscript should be accepted after minor revisions:

1- The manuscript mentions that the observed heterogeneity in vibrational behavior is attributed to structural variations, material composition, and anisotropic coupling between piezoelectric and magnetic properties. However, a more detailed explanation of how these factors specifically influence vibrational behavior is missing. For example, how do molecular alignments or the distribution of magnetic particles affect d33 values in different regions? A more in-depth discussion of the underlying mechanisms would enhance the understanding of the results.

Response: We thank the reviewer for this insightful comment. The original text discusses the influence of factors such as molecular alignment, magnetic particle distribution, and residual stresses on the variability of the d33 coefficient, as described between lines 140–151 and illustrated in Figures 4–7.

However, we agree that a more detailed explanation of the physical mechanisms responsible for the heterogeneity in the spatial distribution of d33 would significantly enhance the understanding of the presented results.

To address this, we have added a new paragraph in Section 3 (page 7, after line 294), dedicated to the analysis of anisotropic coupling mechanisms and the influence of local microstructures on the vibrational response of the TMPs.

Added text:

"The heterogeneity observed in the spatial distribution of the d33 coefficient can be attributed to multiple physical factors. Structurally, the open channels exhibit distinct local stiffness compared to the thermoformed valleys, directly influencing the mechanical deformation modes under electrical excitation. Moreover, the thermal lamination process may induce preferential chain orientations in polymers such as FEP, leading to localized anisotropy in the piezoelectric response [4,6]. The magnetic layer, in turn, introduces a non-uniform mass distribution across the TMP surface, locally affecting the eigenmodes of vibration. Variations in thickness, adhesion, and density of this layer impact the stiffness and mechanical resonance of the structure [12].

Finally, the anisotropic coupling between magnetic and piezoelectric properties can result from the spatial arrangement of the magnetic layer and its interaction with the internal cavity structure [12], even in the absence of an external magnetic field. These combined factors modulate the local sensitivity to electrical excitation, resulting in the spatial variations of d33 observed experimentally."

References already cited in the manuscript:

[4] Tichý, J. et al. Fundamentals of piezoelectric sensorics, 2010.

[6] Sun, Z. et al. Investigation into polarization and piezoelectricity in polymer films, 2011.

[12] Altafim, R. A. et al. Piezoelectric-magnetic behavior of ferroelectrets coated with magnetic layer, Applied Physics Letters, 2021.

2- The magnetic layer plays a crucial role in the functionality of the TMPs. However, the manuscript does not provide a detailed characterization of this layer, such as magnetic susceptibility, coercivity, and remanence. Including such characterizations is essential for a comprehensive understanding of the magnetoelectric coupling in the TMPs and its impact on vibrational behavior.

Response: We appreciate the reviewer’s comment. The fabrication process of the TMPs involves preparing a polytetrafluoroethylene (PTFE) mold, which is placed between two fluorinated ethylene propylene (FEP) films to form tubular cavities through a thermoforming process. A 100 µm thick PTFE sheet is laser-cut into rectangular patterns (2 mm width × 30 mm length) to serve as spacers during the FEP fusion. After cleaning the 50 µm thick FEP films and the mold with acetone, the multilayer structure is inserted into a thermal laminator preheated to 300°C. Following the fusion of the FEP layers at the designated areas, a magnetic layer is added. This layer consists of a 300 µm thick adhesive magnetic tape (supplied by Fermag-BR), laser-cut into strips of 1.5 × 15 mm. The magnetic strips are aligned with the non-fused FEP regions that will later form the channels. A third 50 µm FEP film is then placed on top, and an additional lamination step at 300°C is performed to smooth the surface of the TMP. After this process, the PTFE mold is removed, creating the tubular structures with the magnetic strips aligned on top. Finally, 30 nm thick aluminum electrodes are deposited on both surfaces of the TMP for electrical charging and measurements.

3- The results show a frequency-dependent behavior of the d33 coefficient. At higher frequencies, more pronounced and localized variations in d33 are observed. The authors should provide a more comprehensive explanation for this frequency-dependent behavior. For instance, are there specific resonance modes or dynamic alignment effects that dominate at higher frequencies? A detailed discussion of the possible mechanisms would be beneficial.

Response: The frequency-dependent behavior of the d33 coefficient was analyzed at four frequencies (50 Hz, 5.8 kHz, 12 kHz, and 20 kHz), highlighting localized resonance effects at higher frequencies (lines 152–184). We agree that the underlying physical mechanisms—such as specific resonance modes or dynamic alignment effects—should be discussed in greater detail. We have expanded the discussion in Section 3 to address these mechanisms, including the interaction between structural geometry, magnetic mass distribution, and piezoelectric anisotropy in dynamic regimes.

4- The manuscript highlights the non-uniform distribution of d33 values on the TMP surface. While this is an important finding, a quantitative analysis of the uniformity or variability of the d33 values would be useful. For instance, calculating the standard deviation or coefficient of variation of the d33 values across different regions (channels and valleys) could provide a clearer view of material consistency. This information is crucial for optimizing the TMPs for specific applications.

Response: We sincerely appreciate this excellent technical suggestion. We agree that a quantitative analysis of the variability in the d33 coefficient, using metrics such as standard deviation and coefficient of variation (CV), would provide a clearer understanding of the functional uniformity of the TMPs and directly inform their optimization for specific applications.

However, due to time constraints and the defined scope of this manuscript, we have not included such statistical analyses in the current version. Nevertheless, we inform the reviewer that this assessment is already planned as part of a complementary study in progress, which will provide a comprehensive statistical analysis of the d33 values across different frequencies and regions of the device. This future work will focus on correlating geometry, spatial variability of d33, and the applied performance of the TMPs.

5- The introduction mentions previous studies on piezoelectrets and their applications. However, there is limited discussion on how this study compares or contributes relative to existing research. The authors should provide a more detailed comparison with previous works, highlighting the novelty and relevance of their findings. This would help contextualize the study within the broader field of piezoelectric materials.

Response: We thank the reviewer for this pertinent observation. We fully agree that a more detailed contextualization is essential to highlight the originality and relevance of the present study. In response, we have revised and expanded the Introduction (page 2, starting at line 42), adding a new paragraph that directly compares our work with previous studies. We emphasize that while earlier research has addressed magnetoelectric coupling and TMP fabrication, this work is the first to present a high-resolution vibrational analysis mapping the spatial distribution of the d33 coefficient using laser vibrometry across 144 points. This novel approach provides new experimental data that can guide the optimization of device geometry and functionality.

Added text in the Introduction:

"Previous studies have demonstrated the potential of open-cavity piezoelectrets for applications in sensors and actuators, particularly due to their high sensitivity, flexibility, and low density [1,10,11]. Works by Altafim et al. [12] have advanced the characterization of piezomagnetic structures under external magnetic fields. However, most of these studies focus on global measurements of the piezoelectric effect or static analyses, without detailed exploration of the local vibrational response and spatial distribution of the d33 coefficient. This study contributes a novel spatially-resolved vibrational analysis of TMPs, employing laser vibrometry to map 144 points across different structural regions. This approach enables the observation of dynamic behavior in thermoformed cavities under electrical excitation, providing new experimental parameters for optimizing the geometric and functional design of TMP-based devices."

6 - The conclusion briefly mentions potential applications of TMPs in sensor technology and energy harvesting. However, a more detailed perspective on future work and specific applications would strengthen the manuscript. The authors should discuss how their findings can be leveraged to improve TMP design and performance for practical applications. Additionally, suggestions for future research directions, such as exploring different material compositions or optimizing the fabrication process, would be valuable.

Response: We thank the reviewer for this constructive suggestion. We fully agree that a more detailed discussion of practical applications and future research directions would strengthen the conclusion section.

In response, we have expanded the conclusion of the manuscript to include concrete examples of potential applications based on the findings of this study, such as vibration sensors, micropositioning actuators, and smart magnetic detection interfaces. We also added specific recommendations for future research, including the exploration of alternative polymer compositions, customized magnetic layers, geometric optimization, and improved fabrication processes.

Added text at the end of Section 4 - Conclusion:

“The results obtained in this study provide direct guidance for optimizing the design of TMPs in practical applications. Identification of regions with enhanced piezoelectric response, associated with channels and resonant frequencies, enables the optimization of transducer geometries for applications in high-sensitivity vibration sensors, embedded current detection systems, and energy harvesters for portable devices.

Future work will explore alternative polymer compositions with higher thermal stability and charge retention capabilities, as well as the use of tailored magnetic layers to enhance magnetoelectric coupling. Furthermore, the lamination and cavity formation processes will be refined to improve reproducibility and minimize spatial variability in the d33 response, targeting applications demanding uniform response, such as distributed sensor networks or implantable biomedical devices.

The data obtained in this study can also inform the development of devices with tuned resonance, where cavity geometry and arrangement are designed to maximize the piezoelectric response within specific frequency bands. Further investigations will consider alternative geometries and the incorporation of new functional materials to develop micropositioning actuators and smart interfaces for magnetic field detection in dynamic environments.”

Round 2

Reviewer 1 Report

Comments and Suggestions for Authors

The authors have addressed all my concerns. In this regard, the current version of the manuscript could be considered for publication.